# Cross-modal Learning for Image-Guided Point Cloud Shape Completion

**Emanuele Aiello***
Politecnico di Torino, Italy
emanuele.aiello@polito.it

**Diego Valsesia**
Politecnico di Torino, Italy
diego.valsesia@polito.it

**Enrico Magli**
Politecnico di Torino, Italy
enrico.magli@polito.it

## Abstract

In this paper we explore the recent topic of point cloud completion, guided by an auxiliary image. We show how it is possible to effectively combine the information from the two modalities in a localized latent space, thus avoiding the need for complex point cloud reconstruction methods from single views used by the state-of-the-art. We also investigate a novel weakly-supervised setting where the auxiliary image provides a supervisory signal to the training process by using a differentiable renderer on the completed point cloud to measure fidelity in the image space. Experiments show significant improvements over state-of-the-art supervised methods for both unimodal and multimodal completion. We also show the effectiveness of the weakly-supervised approach which outperforms a number of supervised methods and is competitive with the latest supervised models only exploiting point cloud information.

## 1 Introduction

The rise in popularity of 3D sensing technologies such as depth cameras, laser scanners, LiDARs, etc. is making the processing of point cloud data ever more important. The acquisitions produced by those instruments are often incomplete due to occlusions by objects in the environment, reflections, and viewing angles. This limits the exploitability of those data in tasks like scene understanding [1], robotic vision [2], autonomous driving [3] and many more. Completing a point cloud from partial observations is a challenging ill-posed inverse problem that requires strong prior knowledge about shapes to be effectively regularized.

At the same time, we know that humans are very proficient at mapping the visual concepts learnt from 2D images to understand the 3D world, and are able to successfully infer the shape of partial 3D objects from their 2D experiences. It is thus sensible to expect that point cloud completion techniques can benefit from 2D images to better characterize the 3D shape to be completed. Indeed, several applications of interest in robotic vision can take advantage of multimodal data where the 3D acquisitions of a depth-sensing instrument are paired with images from an RGB camera. It is also worth noting that the two modalities may be acquired from different vantage points, either thanks to disparities in the acquisition geometry or because the vantage point has changed with the passing of time. This makes it clear that the two modalities may carry complementary information and effectively fusing it is key to unlock better completion performance. Nevertheless, the literature on the topic of point cloud completion [4, 5, 6, 7, 8, 9, 10, 11] has largely focused on the single-modality

---

*Code of the project: https://github.com/diegovalsesia/XMFnet

36th Conference on Neural Information Processing Systems (NeurIPS 2022).

problem, where only priors about 3D shapes are exploited. Only recently, image-guided completion has started to receive attention [12].

In this paper, we study how the side information offered by a single image can be used in addition to shape priors to complete a partial point cloud. While following the setting of ViPC [12], we extend the multimodal completion methodology in several different ways. First, ViPC [12] is bottlenecked by the need to estimate a coarse point cloud from the image via single-view reconstruction techniques to fuse the information. We avoid this task by proposing a novel architecture that performs fusion in a latent domain via cross-attention operations on fine-grained, localized representations of the two modalities, coupled with a flexible decoder that allows to complete areas of varied size. Moreover, the multimodal setting is uniquely poised for weakly-supervised learning. In fact, the input image, especially when captured from a different vantage point, may offer a supervisory signal to guide the completion of those areas occluded in the partial point cloud but visible in the image. This is especially interesting for practical applications where it could be difficult to have access to complete shapes, but significantly easier to have images from a different viewing angle. Therefore, we propose to augment the known 3D self-reconstruction losses with the exploitation of a differentiable renderer to measure the fidelity of the completed point cloud in the image space.

Our experiments show that the proposed model significantly outperforms the state-of-the-art on both the supervised and weakly-supervised settings. In particular, the addition of the rendering loss allows the weakly-supervised image-guided model to outperform several supervised baselines and to be competitive with the latest supervised models only exploiting point cloud information.

## 2 Related work

**Point cloud completion**    3D shape completion is a long-standing problem in computer vision. Early works devised explicit geometric descriptors or relied on shape retrieval from large datasets [13] [14] [15] [16]. Since the advent of neural networks operating on raw point cloud data, several models for the completion problem have been studied [17]. They are mostly based on the encoder-decoder architecture, pioneered by PCN [4], which was the first model that did not require any assumption of structure or annotation information about the underlying shape. TopNet [5] presents a hierarchical rooted tree structure that generates structured point clouds as a collection of its subsets. AtlasNet [6] and MSN [18], on the other hand, recreate the point cloud by assessing a set of parametric surface elements. Convolutional-based approaches ([19, 20]) use a voxelixed representation of shapes as input to 3D CNNs; nevertheless, this representation introduces undesirable approximations in the shape due to coordinate quantization effects. GRNet [21] uses techniques to represent point cloud onto a 3D grid, so that CNNs can be exploited, without losing structural information. Recently, VRCNet [8] has proposed a dual path architecture and a VAE-based relation enhancement module for probabilistic modeling. Architectures based on transformers have also been proposed. PointTr [9] changes the transformer block to take advantage of the inductive bias of 3D geometries, creating a geometry-aware block that models local geometry relations. Moreover, SnowflakeNet [22] generates child points by gradually splitting parent points by means of a Skip-Transformer that learns the appropriate splitting modes for particular regions. As a result, the network is able to predict highly detailed shape geometries. Finally, it is worth mentioning that the point cloud completion literature is split between two settings: one, as in the aforementioned works, where the partial input has the same number of points as the completed point cloud and another where the completed point cloud has more points than the input such as in [23, 24]. In this paper, we will consider the former setting.

**View-guided completion**    Recently, the usage of auxiliary data to complement point cloud completion has been introduced by ViPC [12]. The idea is to help the reconstruction objective using side information available as a different imaging modality. In particular, ViPC assumes that an image corresponding to a view of the same object is also available for the completion task, and it exploits the image to retrieve the global shape information that is lacking in the incomplete point cloud. The image is processed by a pre-trained single-view reconstruction model, which estimates a coarse point cloud from the image, representing the entire shape. The key challenge in this setting is how to effectively combine features extracted from the two modalities. Unlike ViPC, our approach leverages direct fusion at a feature level, avoiding the need to explicitly reconstruct a coarse point cloud from a single image, a generally hard inverse problem in itself and full of pitfalls.

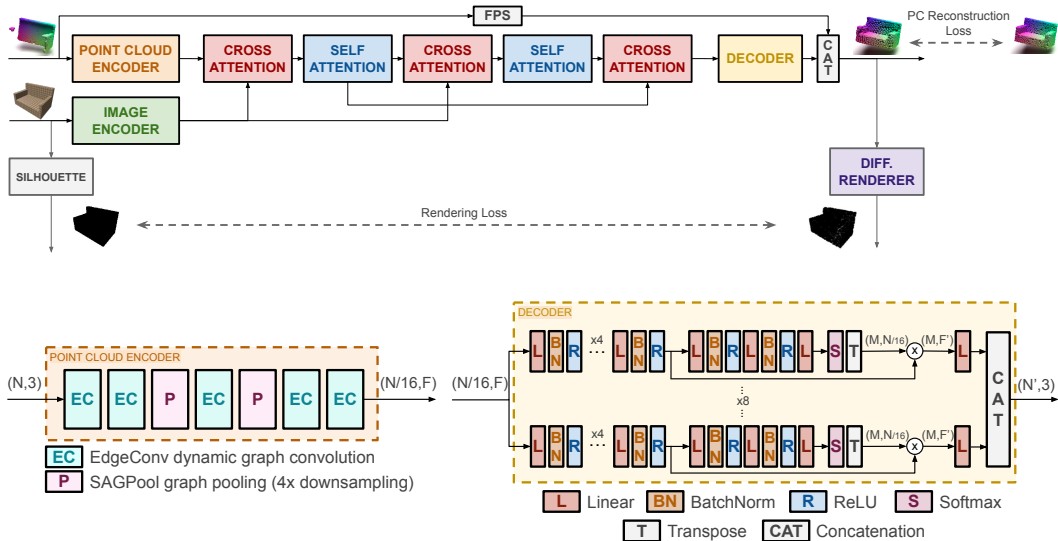

Figure 1: Architecture overview. Localized features from the partial point cloud and the input image are jointly processed via cross- and self-attentions. A decoder reconstructs the target number of points from the feature space with attention-based upsampling. The input partial point cloud is downsampled with farthest point sampling (FPS) and concatenated. Supervised training only uses the point cloud reconstruction loss with respect to the complete point cloud. Weakly-supervised training has a point cloud reconstruction loss with respect to a less partial point cloud and a rendering loss. $N$ is the number of points of the input point cloud; $M$ is the number of points generated by each branch of the decoder, while $F$ represents the feature dimension.

**Self-supervised strategies for completion**   All the previously mentioned approaches rely on complete ground truth as a supervisory training signal. This may be difficult to obtain in real-word scenarios. Self-supervised training strategies avoid the need for retrieving such expensive ground truths. However, the amount of work on this topic is rather limited. Wang et al. [10] use resampling that removes further points from an already partial point cloud and mixup among shapes to train their completion network in a self-supervised manner. Similarly, Mittal et al. [11] also propose an inpainting procedure that leverages further partializations of partial point clouds. To the best of our knowledge, there is no work exploring the availability of a different modality, namely an image to provide a weak supervisory signal to the point cloud completion task.

## 3   Proposed Methods

We address the setting in which a partial point cloud needs to be completed with the assistance of an image of the object taken from a certain viewpoint. Our goal is to study how to leverage this side information in the most effective manner. To this purpose we study i) a supervised learning setting, for which we show that the image features can be effectively fused with those of the partial point cloud in a latent space; ii) a weakly-supervised setting based on the idea that the image may contain clues about the missing part and can thus serve as a supervisory signal. Fig. 1 shows an overview of the proposed method, named XMFnet (Cross-Modal Fusion network) which will be detailed in the next sections.

### 3.1   Architecture and Supervised Setting

At a high level, the architecture of XMFnet is composed by two modality-specific feature extractors that capture localized features of the input point cloud and image, summarized at a small number of points/pixels, followed by a sequence of cross-attention and self-attention operations that progressively merge the two feature spaces. Finally, a decoder upsamples this localized information to estimate a predefined number of points of the missing component. More formally, we denote the partial point cloud as $\mathbf{X} \in \mathbb{R}^{N \times 3}$, the input view as an image $\mathbf{I} \in \mathbb{R}^{P_x \times P_y \times 3}$ and the complete point cloud as $\mathbf{Y} \in \mathbb{R}^{N \times 3}$. The task of our model is to predict a complete shape $\hat{\mathbf{Y}} \in \mathbb{R}^{N \times 3}$ given $\mathbf{X}$ and $\mathbf{I}$ as inputs.

Notice we follow the conventional setting where the partial and complete point clouds have the same number of points, meaning that there is a resampling of the known part.

### 3.1.1 Point Cloud and Image Encoder

The point cloud encoder should extract localized features from the partial shape $\mathbf{X}$. It is important to keep a degree of locality, i.e., associating features to a small number of points $N_X < N$ rather than a single global embedding because the information about the missing part that needs to be estimated by the entire model is also mostly localized. However, it is also important to have a sufficiently large receptive field to infer some global information about the object. For this reason, we adopt a graph-convolutional architecture with graph pooling operations. The architecture is a sequence of graph-convolutional layers (EdgeConv [25]) interleaved by pooling operations (Self-Attention Graph Pooling [26]) to reduce the cardinality of the point cloud. Pooling has the double purpose of expanding the receptive field to also include more global information and reduce the complexity of the subsequent cross-attention operations fusing the two modalities.

Any network that extracts features from an image can be utilized as encoder for view $\mathbf{I}$. The design principles follow those of the point-cloud encoder, i.e., features localized at a subset $N_I < P_x P_y$ of the image pixels obtained from a sufficiently large receptive field are produced as output.

We will refer to the features produced by the point cloud encoder as $\mathbf{H}_X \in \mathbb{R}^{N_X \times F_X}$ and by the image encoder as $\mathbf{H}_I \in \mathbb{R}^{N_I \times F_I}$.

### 3.1.2 Modality Fusion

Once we have collected localized information from the two modalities, we need to combine them effectively to capture their complementary information, despite the obvious domain gap. The attention mechanism is particularly suited to find correspondences between the features of a region of the point cloud and a region of the image. The cross-attention layer in our architecture uses the Transformer's multihead attention mechanism [27]. The point cloud features are projected to form the query tensor, while the image features are projected to form the key and value tensors, and then attention mechanism aggregates the features from different image regions according to the weights determined by the cross-correlation between the two modalities. More formally:

$$\mathbf{Q}_X = \mathbf{H}_X \mathbf{W}_Q, \quad \mathbf{K}_I = \mathbf{H}_I \mathbf{W}_K, \quad \mathbf{V}_I = \mathbf{H}_I \mathbf{W}_V \tag{1}$$

$$\mathbf{H}_{\text{fused}} = \text{softmax} \left( \frac{\mathbf{Q}\mathbf{K}^T}{\sqrt{F}} \right) \mathbf{V} \tag{2}$$

being $\mathbf{W}_Q \in \mathbb{R}^{F_X \times F}$, $\mathbf{W}_K, \mathbf{W}_V \in \mathbb{R}^{F_I \times F}$ the projection weights. The fused features $\mathbf{H}_{\text{fused}} \in \mathbb{R}^{N_X \times F}$ produced by the cross-attention mechanism can be regarded as the original point cloud features enriched by the image features.

The XMFnet architecture depicted in Fig. 1 shows a self-attention layer after the cross-attention fusion. The goal of this operation is to have a permutation-invariant transformation of the features with a global receptive field so that any information from the image not properly integrated can be rectified. Self-attention works exactly like Eq. (2) except for the fact that $\mathbf{Q}, \mathbf{K}, \mathbf{V}$ are all different projections of the same features. Furthermore, a sequence of multiple cross- and self-attention layers can be used to more effectively integrate the information from the two modalities via a "slow" fusion. We remark that at the end of this sequence we use a special cross-attention layer that merges information from the end and the beginning of the sequence allowing better flexibility in the decision of the desired abstraction level (higher-level features cross-attend lower-level features).

### 3.1.3 Decoder and Supervised Loss

The decoder is a crucial component of our architecture as it should take the joint feature embedding and learn to reconstruct a complete point cloud preserving both global and local structure. To be precise, the decoder seeks to estimate the positions of a number of points that upper bounds the size of the missing part, so that they can be concatenated to a version of the input partial point cloud subsampled by means of farthest point sampling (FPS). This mechanism is reminiscent of what is done in the setting with a variable number of points where only the missing part is estimated [23, 24].

For example, in our experiments, we upper bounded the size of the missing part to $50\%$ of the total number of points, thus having $N'$ points estimated by the decoder concatenated to $N'$ points from the subsampled input. However, our method allows to be flexible and handle more incomplete inputs by simply tuning the desired ratio of points to be estimated and points provided from the partial input. Typically, the latent space where we perform feature fusion is much more localized to constrain complexity and allow higher-level features so that $N_X \ll N'$, thus requiring the decoder to upsample the feature field. We perform this operation by using a number of attention-based operations, inspired by the work in [28], that convert features to points in parallel with the idea that each branch specializes on the reconstruction of a sub-region of the missing part. The structure is depicted in Fig.1. More formally, calling $\mathbf{H} \in \mathbb{R}^{N_X \times F}$ the features provided to the decoder, the output $\hat{\mathbf{Y}}_i \in \mathbb{R}^{N'/K \times 3}$ of each of the $K$ branches is computed as:

$$\mathbf{Z}_i = \mathrm{MLP}_i^{\mathrm{proj}}(\mathbf{H}) \qquad i = 1, \ldots, K \tag{3}$$

$$\hat{\mathbf{Y}}_i = \left(\mathrm{softmax}(\mathrm{MLP}_i^{\mathrm{dec}}(\mathbf{Z}_i))^T \mathbf{Z}_i\right) \mathbf{W}_{\mathrm{out},i} \qquad i = 1, \ldots, K \tag{4}$$

where $\mathrm{MLP}_i^{\mathrm{proj}} : \mathbb{R}^F \to \mathbb{R}^{F'}$, $\mathrm{MLP}_i^{\mathrm{dec}} : \mathbb{R}^{F'} \to \mathbb{R}^{N'/K}$ are multilayer perceptrons with different weights for each branch, projecting features to $K$ subspaces and generating attention weights for the resampling process, respectively. $\mathbf{W}_{\mathrm{out},i} \in \mathbb{R}^{F' \times 3}$ is projection matrix to 3D space. Finally, the completed point cloud is generated by concatenation of the outputs of all decoder branches and the partial input subsampled by FPS as:

$$\hat{\mathbf{Y}} = \left[\hat{\mathbf{Y}}_1, \hat{\mathbf{Y}}_2, \ldots, \hat{\mathbf{Y}}_K, \mathrm{FPS}(\mathbf{X})\right]. \tag{5}$$

Supervised training is performed using the L1 Chamfer Distance (CD) between the generated shapes and the ground truth shapes, defined as follows:

$$\mathcal{L}_{\mathrm{CD}}(\mathbf{Y}, \hat{\mathbf{Y}}) = \frac{1}{2N} \sum_{\mathbf{y} \in \mathbf{Y}} \min_{\hat{\mathbf{y}} \in \hat{\mathbf{Y}}} \|\mathbf{y} - \hat{\mathbf{y}}\| + \frac{1}{2N} \sum_{\hat{\mathbf{y}} \in \hat{\mathbf{Y}}} \min_{\mathbf{y} \in \mathbf{Y}} \|\hat{\mathbf{y}} - \mathbf{y}\|. \tag{6}$$

### 3.2 Weakly-supervised Setting

The multimodal completion problem addressed in this paper is uniquely poised for weakly-supervised learning in a setting where the full ground truth point cloud is not available. In fact, the image available as input may contain complementary information with respect to the point cloud and, crucially, cues about the missing part. This is especially true if the image is collected from a different viewpoint or at a different time with respect to the point cloud, resulting in different kinds of occlusions.

Existing architectures for self-supervised completion [10] [11] rely solely on point cloud supervision, due to their unimodal nature. The key insight of the proposed method (Fig. 2) is to supplement completion losses on points with a loss measuring a reprojection error in the image space. In particular, we measure whether the reconstructed point cloud produced by the architecture described in Sec. 3.1 leads to an image similar to the input one, when captured from the correct viewpoint.

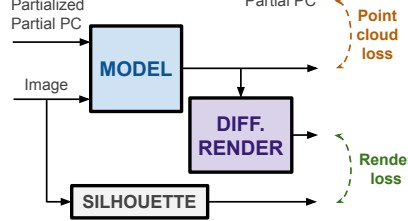

Figure 2: Weakly-supervised training.

In order to measure this information and use it in the training process, we include a differentiable rendering module based on alpha compositing [29] which generates a rasterized version of the object, employing provided camera parameters. In order to ensure consistency with the input image, intrinsic and extrinsic camera parameters may be estimated with a number of well-known methods [30] [31] [32]. In order to minimize the domain shift between the input image and the result of the rendering process, we work on silhouettes, i.e., binary masks of objects. The differentiable renderer produces a soft silhouette with continuous values, while the input image is directly binarized. Inevitable inaccuracies in the camera parameters and rendering process will typically yield unreliable borders of the silhouette. Therefore, we also compute a border mask with a simple edge detector (Laplacian of

Gaussian) and discount the loss function by a factor $\varepsilon < 1$ for the pixels in the mask. In summary, our rendering loss is defined as:

$$\mathcal{L}_{\text{render}} = \left\| \mathbf{M} \odot \left[ R(\hat{\mathbf{Y}}) - S(\mathbf{I}) \right] \right\|_1, \qquad \mathbf{M}_{i,j} = \begin{cases} \varepsilon & \text{if } (i,j) \in \text{edge} \\ 1 & \text{otherwise} \end{cases} \qquad (7)$$

being $R$ the differentiable silhouette renderer and $S$ the silhouette binarizer. We remark that [33] proposes to use rendering to improve performance in point cloud completion, aiding the learning process through an image domain supervision. However, their approach is supervised and is based on rendering the ground truth and generated point cloud in depth-maps with different view-points.

In addition to the rendering loss, our weakly-supervised framework also adopts self-supervision in the point cloud domain. In particular, we use a combination of the resampling and mixup approaches proposed in [10, 11]. Resampling consists in removing random portions of the original partial input point cloud, yielding even more partial shapes. As a result, the original partial input is employed as a pseudo-ground truth. Mixup combines a pair of partial shapes weighed according to Beta distribution in an attempt at increasing the complexity of the shapes processed by the network. Differently from [10], we also have images associated with a partial shape in our setting. Hence, we also mix the images in such a way that the mixup technique is carried out symmetrically for the two modalities.

We remark that the rendering loss is comparatively weaker than the point cloud loss and, by itself, has a number of ambiguities due to the lack of depth information and the use of silhouettes. For this reason, it is important to combine it with the point cloud loss. We found that the density-aware Chamfer Distance (DCD) [34], a version of CD that is more sensitive to non-uniform point distributions is superior in this weakly-supervised scenario to regularize the ambiguities of the rendering loss. Our overall weakly-supervised training procedure is therefore as follows. We alternate between a step that optimizes the point cloud loss consisting in a weighted CD:

$$\mathcal{L}_{\text{PC}} = \frac{1-\beta}{2N} \sum_{\mathbf{y} \in \mathbf{Y}} \min_{\hat{\mathbf{y}} \in \hat{\mathbf{Y}}} \|\mathbf{y} - \hat{\mathbf{y}}\| + \frac{\beta}{2N} \sum_{\hat{\mathbf{y}} \in \hat{\mathbf{Y}}} \min_{\mathbf{y} \in \mathbf{Y}} \|\hat{\mathbf{y}} - \mathbf{y}\|, \qquad (8)$$

and a step optimizing a combination of DCD and rendering loss:

$$\mathcal{L}_{\text{I}} = \left[ \frac{1}{2N} \sum_{\mathbf{y} \in \mathbf{Y}} \left( 1 - \frac{1}{N} e^{-\alpha \|\mathbf{y} - \mathbf{w}\|_2} \right) + \frac{1}{2N} \sum_{\hat{\mathbf{y}} \in \hat{\mathbf{Y}}} \left( 1 - \frac{1}{N} e^{-\alpha \|\hat{\mathbf{y}} - \mathbf{z}\|_2} \right) \right] + \lambda \mathcal{L}_{\text{render}} \qquad (9)$$

where $\mathbf{w} = \arg\min_{\hat{\mathbf{y}} \in \hat{\mathbf{Y}}} \|\mathbf{y} - \hat{\mathbf{y}}\|_2$ and $\mathbf{z} = \arg\min_{\mathbf{y} \in \mathbf{Y}} \|\mathbf{y} - \hat{\mathbf{y}}\|_2$. Notice that $\mathcal{L}_{\text{PC}}$ and the DCD part of $\mathcal{L}_{\text{I}}$ use resampling and mixup for point clouds, while $\mathcal{L}_{\text{render}}$ does not ($\mathcal{L}_{\text{I}}$ is computed with full minibatch for the point-cloud part and half minibatch with the original partials for rendering).

## 4 Experimental results

### 4.1 Experimental Settings and Implementation Details

All the experiments are conducted on the ShapeNet-ViPC[12] dataset. The dataset contains 38,328 objects from 13 categories; for each object it comprises 24 partial point clouds with occlusions generated under 24 viewpoints, using the same settings as ShapeNetRendering [35]. The input and ground truth point clouds contains $N = 2048$ points each. Each 3D shape is rotated to the pose corresponding to a certain view point after being normalized within the bounding sphere with radius of 1. Images are generated from the 24 view points of ShapeNetRendering and have a resolution of $224 \times 224$ pixels. For all the experiments in this paper, we employ the same selection used in [12]: we used 31,650 objects from eight categories, with $80\%$ of them for training and $20\%$ for testing.

The partial point cloud is downsampled by farthest point sampling to $N' = 1024$ points and concatenated to the output of the decoder that produces $N' = 1024$ points leading to a completed point cloud with 2048 points. The decoder has $K = 8$ branches, each of them producing $M = 128$ points. The point cloud encoder employs EdgeConv and SAGPooling layers; the EdgeConv layers selects $k = 20$ nearest neighbors, while the two pooling layers use $k = 16$ and $k = 6$ nearest neighbors, respectively. The original point cloud is overall downsampled by a factor of 16, resulting in $N_X = 128$ points with $F_X = 256$ features. The image encoder is built with a ResNet18 [36] as

Table 1: Mean Chamfer Distance per point ($\times 10^{-3}$). ShapeNet-ViPC dataset, supervised.

| Methods | Avg | Airplane | Cabinet | Car | Chair | Lamp | Sofa | Table | Watercraft |
|---|---|---|---|---|---|---|---|---|---|
| AtlasNet [6] | 6.062 | 5.032 | 6.414 | 4.868 | 8.161 | 7.182 | 6.023 | 6.561 | 4.261 |
| FoldingNet [39] | 6.271 | 5.242 | 6.958 | 5.307 | 8.823 | 6.504 | 6.368 | 7.080 | 3.882 |
| PCN [4] | 5.619 | 4.246 | 6.409 | 4.840 | 7.441 | 6.331 | 5.668 | 6.508 | 3.510 |
| TopNet [5] | 4.976 | 3.710 | 5.629 | 4.530 | 6.391 | 5.547 | 5.281 | 5.381 | 3.350 |
| ECG [7] | 4.957 | 2.952 | 6.721 | 5.243 | 5.867 | 4.602 | 6.813 | 4.332 | 3.127 |
| VRC-Net [8] | 4.598 | 2.813 | 6.108 | 4.932 | 5.342 | 4.103 | 6.614 | 3.953 | 2.925 |
| ViPC [12] | 3.308 | 1.760 | 4.558 | 3.183 | 2.476 | 2.867 | 4.481 | 4.990 | 2.197 |
| **XMFnet** | **1.443** | **0.572** | **1.980** | **1.754** | **1.403** | **1.810** | **1.702** | **1.386** | **0.945** |

Table 2: Mean F-Score @ 0.001. ShapeNet-ViPC dataset, supervised

| Methods | Avg | Airplane | Cabinet | Car | Chair | Lamp | Sofa | Table | Watercraft |
|---|---|---|---|---|---|---|---|---|---|
| AtlasNet [6] | 0.410 | 0.509 | 0.304 | 0.379 | 0.326 | 0.426 | 0.318 | 0469 | 0.551 |
| FoldingNet [39] | 0.331 | 0.432 | 0.237 | 0.300 | 0.204 | 0.360 | 0.249 | 0.351 | 0.518 |
| PCN [4] | 0.407 | 0.578 | 0.270 | 0.331 | 0.323 | 0.456 | 0.293 | 0.431 | 0.577 |
| TopNet [5] | 0.467 | 0.593 | 0.358 | 0.405 | 0.388 | 0.491 | 0.361 | 0.528 | 0.615 |
| ECG [7] | 0.704 | 0.880 | 0.542 | 0.713 | 0.671 | 0.689 | 0.534 | 0.792 | 0.810 |
| VRC-Net [8] | 0.764 | 0.902 | 0.621 | 0.753 | 0.722 | **0.823** | 0.654 | 0.810 | 0.832 |
| ViPC [12] | 0.591 | 0.803 | 0.451 | 0.512 | 0.529 | 0.706 | 0.434 | 0.594 | 0.730 |
| **XMFnet** | **0.796** | **0.961** | **0.662** | **0.691** | **0.809** | 0.792 | **0.723** | **0.830** | **0.901** |

backbone, it extracts $N_I = 14 \times 14 = 196$ pixels with $F_I = 256$ features. The multihead attention has 4 attention heads, with embedding size $F = 256$. In the $\mathcal{L}_I$ loss we use $\lambda = 0.15$. The mask factor for the edge detector has been set to $\varepsilon = 0.4$.

The differentiable renderer has been implemented with PyTorch3D[37]. The rendered silhouettes $H \times W$ has size $224 \times 224$ that is the same size of input views in our experiments. We adopt radius $\rho = 0.025$ in point rasterization. The proposed framework is implemented in PyTorch and trained on an Nvidia V100 GPU. Class-specific training is performed for all models, using the Adam optimizer [38] for roughly 200 epochs with a batch size of 128. The learning rate is initialized to 0.001 and reduced by a factor of 10 at epoch 25 and 125.

## 4.2 Main Results

### 4.2.1 Supervised Learning

We first compare XMFnet against several baselines for the point cloud completion under supervised learning. Since the new multimodal setting with an auxiliary image has been introduced only recently, ViPC [12] represents the only method fully comparable to ours. However, we also report the results of a number of state-of-the-art architectures for completion with only point cloud input, when retrained on the ViPC dataset. AtlasNet[6] reconstructs a point cloud by estimating parametric surface elements. FoldingNet[39] is a 2-D grid based auto-encoder. PCN[4] is an encoder-decoder framework that reconstructs the point cloud in a coarse-to-fine manner. TopNet[5] has a rooted tree structure in the decoder. ECG[7] is an edge-aware completion method based on Graph Convolutions. VRC-Net[8] is the most recent method adopting a VAE-based model with a dual path architecture and probabilistic modeling. In line with the previous evaluation protocols, we use CD and F-score [40] as metrics for the reconstruction quality. Before evaluating the CD, we normalize the output of the models to fit into the unit sphere. Table 1 and Table 2 report the experimental results and show XMFnet outperforming the other techniques by a significant margin. While part of this gain with respect to state-of-the-art models for point cloud completion can be attributed to the use of the input image, it is worth noting that we also report significant improvements over the multimodal ViPC. This highlights the sub-optimality of performing modality fusion by resorting to estimating a coarse point cloud from the single input image, as in ViPC, rather than working in a latent feature space. Qualitative

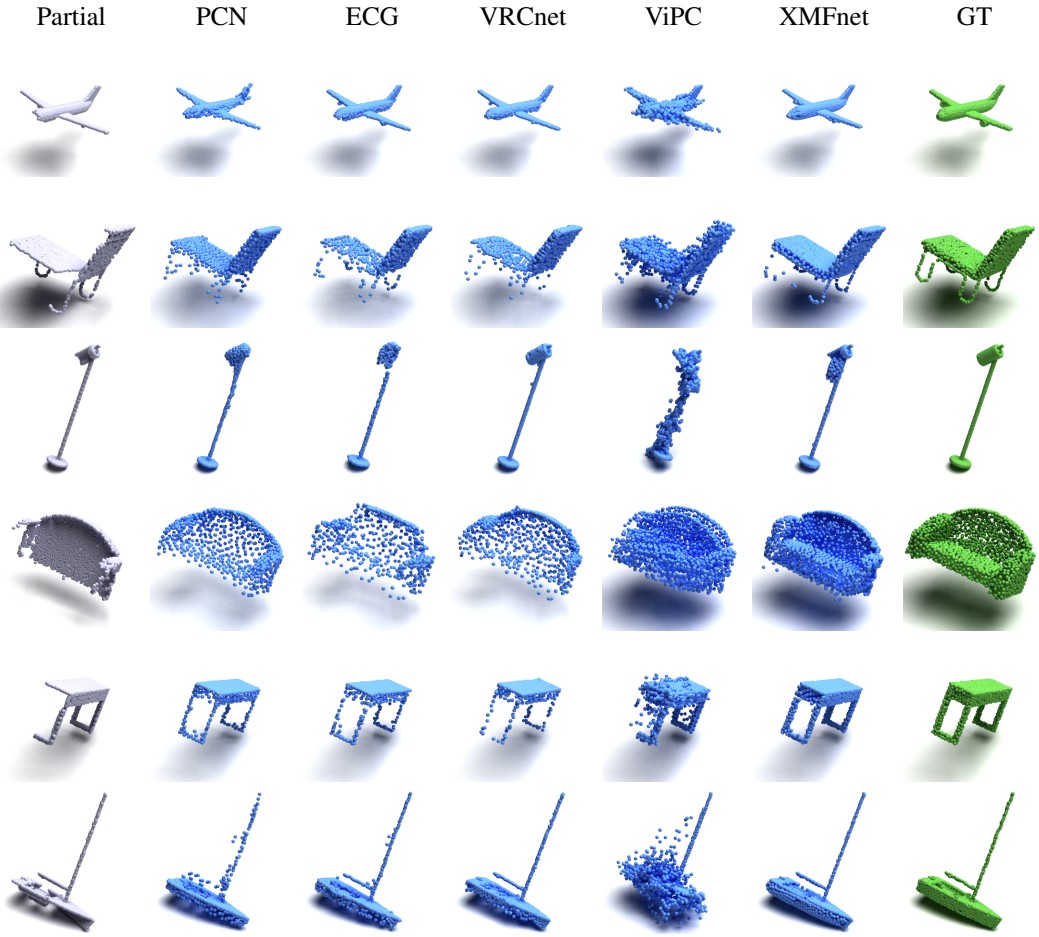

| Partial | PCN | ECG | VRCnet | ViPC | XMFnet | GT |

Figure 3: Qualitative comparison of completed point clouds for different classes.

comparisons[2] are shown in Fig. 3. Our method is capable of producing cleaner completions than the other baselines, with fewer outliers and a more uniform point distribution.

### 4.2.2 Weakly-supervised

We remark that we are the first to propose a weakly-supervised training strategy for multimodal completion, so the setting in which supervisory information can be gathered from an input image is unexplored. For this reason, we compare to a number of unimodal and multimodal supervised methods as well as a self-supervised version using resampling and mixup of a unimodal state-of-the-art model[3]. The results are reported in Table 3 for the CD and Table 4 for the F-Score. We notice that our weakly-supervised method outperforms a number of supervised baselines and it is close to the performance of the most recent unimodal supervised models.

### 4.3 Ablation Studies

In order to verify the effectiveness of the proposed design, we study the impact of the auxiliary image input on the completion performance. To ensure a comparison as fair as possible, the version of XMFnet that uses only point cloud information has the image encoder removed and the cross-attention blocks replaced with self-attention ones.

---

[2]Visualizations for ViPC [12] have been kindly provided by the original authors.

[3]We remark the difficulty in reproducing several published methods in the self-supervised setting. The code for [10, 11] was not available. The code to reproduce ViPC [12] is also incomplete so we cannot retrain the most sensible self-supervised baseline of ViPC + Resampling + Mixup.

Table 3: Mean Chamfer Distance per point $(\times 10^{-3})$. ShapeNet-ViPC dataset.

| Methods | Airplane | Lamp | Watercraft |
|---|---|---|---|
| AtlasNet [6] | 5.032 | 7.182 | 4.261 |
| FoldingNet [39] | 6.504 | 6.368 | 3.882 |
| PCN [4] | 4.246 | 6.331 | 3.510 |
| TopNet [5] | 3.710 | 5.547 | 3.350 |
| ECG [7] | 2.952 | 4.602 | 3.127 |
| VRC-Net [8] | 2.813 | 4.103 | 2.925 |
| VRC-Net(self-sup.) | 4.315 | 8.023 | 7.259 |
| ViPC [12] | 1.760 | 2.867 | 2.197 |
| XMFnet (sup.) | 0.572 | 1.810 | 0.945 |
| **XMFnet (weakly-sup.)** | 2.426 | 6.269 | 3.423 |

Table 4: Mean F-Score @ 0.001. ShapeNet-ViPC dataset.

| Methods | Airplane | Lamp | Watercraft |
|---|---|---|---|
| AtlasNet [6] | 0.509 | 0.426 | 0.551 |
| FoldingNet [39] | 0.432 | 0.360 | 0.518 |
| PCN [4] | 0.578 | 0.456 | 0.577 |
| TopNet [5] | 0.593 | 0.491 | 0.615 |
| ECG [7] | 0.880 | 0.689 | 0.810 |
| VRC-Net [8] | 0.902 | 0.823 | 0.832 |
| VRC-Net(self-sup.) | 0.689 | 0.710 | 0.673 |
| ViPC [12] | 0.803 | 0.706 | 0.730 |
| XMFnet (sup.) | 0.961 | 0.792 | 0.901 |
| **XMFnet (weakly-sup.)** | 0.742 | 0.542 | 0.704 |

Table 5: Unimodal vs. Multimodal completion (supervised)

| Method | Avg | Airplane | Cabinet | Lamp |
|---|---|---|---|---|
| Unimodal | 1.570 | 0.626 | 2.114 | 1.980 |
| **Multimodal** | **1.470** | **0.572** | **1.973** | **1.810** |
| ↳ **best view** | 1.223 | 0.545 | 1.426 | 1.697 |
| ↳ **worst view** | 1.819 | 0.722 | 2.621 | 2.115 |

Table 6: Ablation study for the Weakly-Supervised method (*airplane*)

| Resampling | Mixup | Rendering | CD($10^{-3}$) |
|---|---|---|---|
| ✓ | ✗ | ✗ | 4.568 |
| ✓ | ✓ | ✗ | 4.239 |
| ✓ | ✓ | ✓ | **2.426** |

Table 7: Ablation study for the Weakly-Supervised method - DCD (*cabinet*)

| DCD | CD($10^{-3}$) |
|---|---|
| ✗ | 3.012 |
| ✓ | 2.426 |

The unimodal architecture is then trained with the same settings as the multimodal one, and the results are reported in Table 5. The results show that the addition of the image input provides a significant improvement in performance. Notice that besides the result averaged over all the possible 24 views, we also report the performance with the worst view and the best view. Indeed, we are interested in investigating how the viewpoint of the image affects completion.

Fig. 4 reports the average CD for different views, ordered from worst to best, for the cabinet category. It is clear that some views provide complementary information due to their different vantage point and allow to substantially improve over the average result. A small number of "bad" views leads to results comparable to the unimodal case.

Furthermore, it is interesting to study the impact of our novel rendering loss in the weakly-supervised setting. We noticed that it allows the training process to have a faster and smoother convergence and that the overall completion performance is increased from both a qualitative and quantitative point of view. A qualitative comparison between the weakly-supervised strategy with and without the rendering module can be visualized in Fig. 5 and quantitative

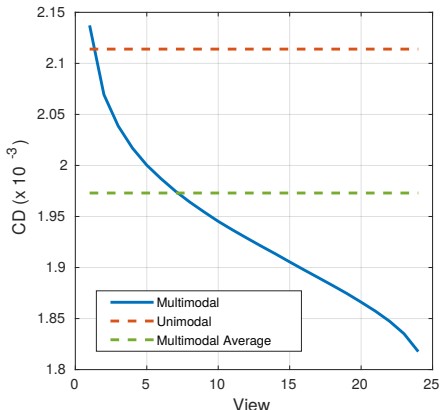

Figure 4: Impact of image contribution as function of point of view, sorted by reconstruction CD (from worst to best) averaged over cabinet category, supervised setting.

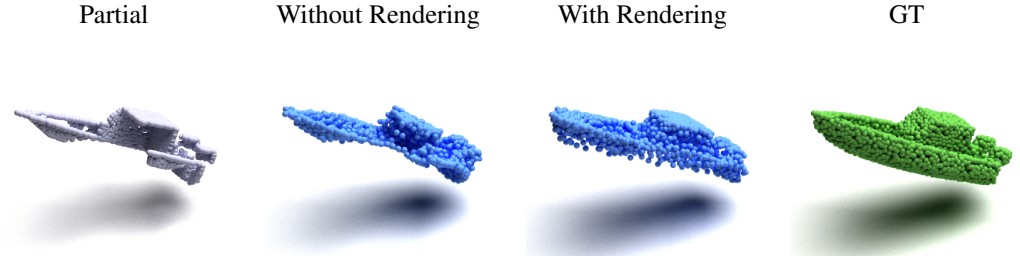

Partial      Without Rendering      With Rendering      GT

Figure 5: Qualitative visualization of the effect of the proposed weakly-supervised rendering loss. The sample without rendering has CD = 7.812, the one with rendering has CD= 3.743.

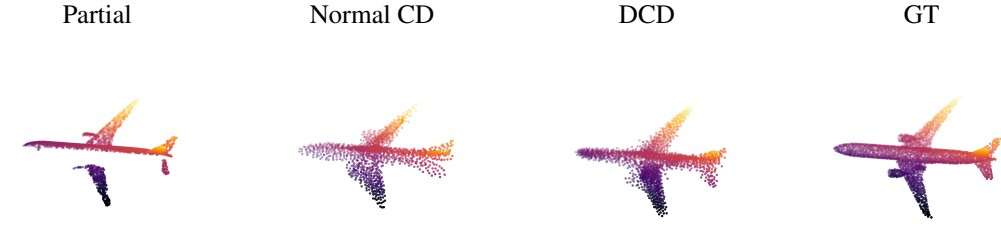

Partial      Normal CD      DCD      GT

Figure 6: Qualitative visualization of the effect of the DCD for the weakly-supervised setting.

results are reported in Table 6. The mixup loss provides only a small improvement from the perspective of the CD. However, it substantially improves the completed shape from a qualitative point of view, helping the network generate more complete shapes. Moreover, the computational overhead due to creating the mixed input shape is very small, so we decided to keep the method as it offered a very favorable cost-performance trade-off. We also include the ablation for the DCD component of the weakly-supervised loss, we found it helpful from both a qualitative and quantitative point of view. Table 7 reports the impact of the DCD on the *cabinet* category, while Fig. 6 provides a qualitative example, where the shape completed with the DCD presents a more uniform distribution of points and a better overall quality.

## 5    Conclusions

In this paper, we explored the topic of point cloud completion guided by an auxiliary image, discovering that effective fusion can be achieved in a latent space via cross-attention. Our method achieves state-of-the-art results on the ShapeNetViPC-Dataset. Moreover, we showed how this setting lends itself to weakly-supervised learning where the image can be used for supervision via a differentiable rendering approach, when the full ground truth point cloud is not available. The major limitation of our work is the lack of study of a real-world scenario for the proposed framework. In future work, we will focus on extending the work to real acquisitions, thus dealing with complex effects like acquisition noise, background or additional occlusions in the auxiliary images, and many more. This paper has provided a proof of concept that effective multimodal completion is possible but a more in-depth study of such issues on real scenes is needed, along with suitable improvements to our design towards increased robustness.

## Acknowledgments and Disclosure of Funding

Computational resources were provided by HPC@POLITO, a project of Academic Computing within the Department of Control and Computer Engineering at the Politecnico di Torino (http://www.hpc.polito.it). This research received no external funding.

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
