# Cross-modal Learning for Image-Guided Point Cloud Shape Completion - Supplementary material

**Emanuele Aiello**[*]
Politecnico di Torino, Italy
emanuele.aiello@polito.it

**Diego Valsesia**
Politecnico di Torino, Italy
diego.valsesia@polito.it

**Enrico Magli**
Politecnico di Torino, Italy
enrico.magli@polito.it

## 1 Resource Usage Comparison

Table 1: Computational Comparison

| Methods | #Params (M) | Inference Time (ms) |
|---|---|---|
| PCN [2] | 6.86 | 2.7 |
| VRC-Net [1] | 17.47 | 183.3 |
| ViPC [3] | 11.48 | 62.9 |
| XMFnet | 10.04 | 16.2 |

We evaluated the resource usage by PNC [2], VRC-Net [1], ViPC [3] and our XMFnet. The results are reported in Table 1, our model has a lower number of parameters and it is faster in inference with respect to the state of the art ViPC and VRC-Net. This is due to the good parameters' exploitation of our architecture and the fact that differently from ViPC we do not reconstruct a coarse point cloud from the image, avoiding unnecessary computational overhead.

## 2 Standard Deviation of Evaluation

Table 2: Mean Chamfer Distance per point ($\times 10^{-3}$). ShapeNet-ViPC dataset. Standard deviation for each category. XFMnet.

|  | Airplane | Cabinet | Car | Chair | Lamp | Sofa | Table | Watercraft |
|---|---|---|---|---|---|---|---|---|
| CD | 0.572 | 1.980 | 1.754 | 1.403 | 1.810 | 1.702 | 1.386 | 0.945 |
| std | ±0.037 | ±0.066 | ±0.075 | ±0.064 | ±0.138 | ±0.078 | ±0.055 | ±0.042 |

The value of standard deviation for each category are reported in Table 5. Lamp category is the one with higher variability and is also the one with the lowest F-Score.

---

[*]Code of the project: https://github.com/diegovalsesia/XMFnet

36th Conference on Neural Information Processing Systems (NeurIPS 2022).

# 3   Completion results as function of input view

We show how different input views affect the completion in Figure 1. We generated several completion starting from the same partial input point cloud and different input views. It can be noticed that views that contain more information about the missing regions provide better completion results.

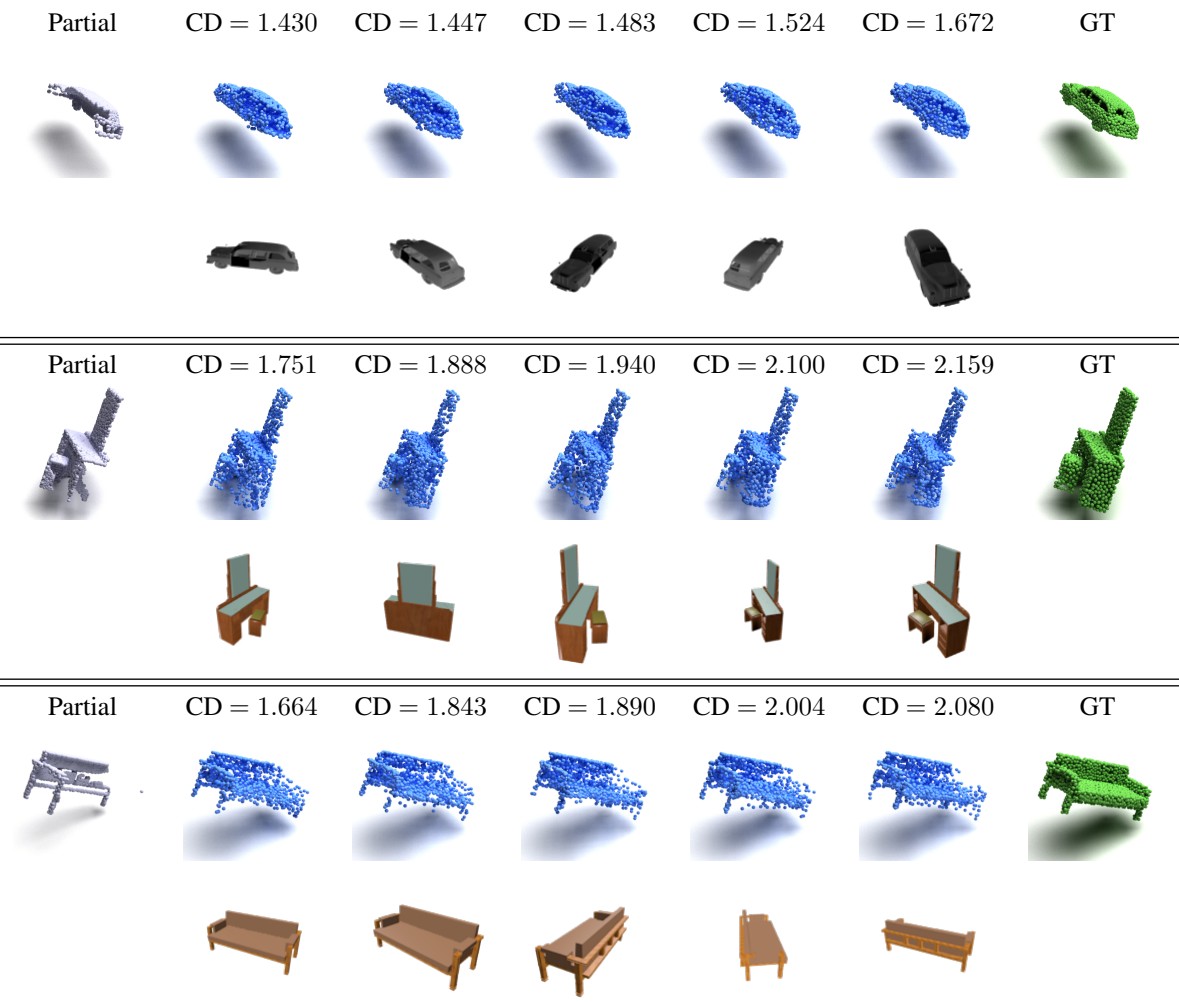

Figure 1: Completion Results with respect to different input views.

# 4 Failure Cases

We show some difficult samples where our model struggles to reconstruct one particular challenging class is the lamp category. Figure 2 shows difficult samples for the supervised setting, while Figure 3 for the self-supervised one.

Partial             Reconstructed             GT

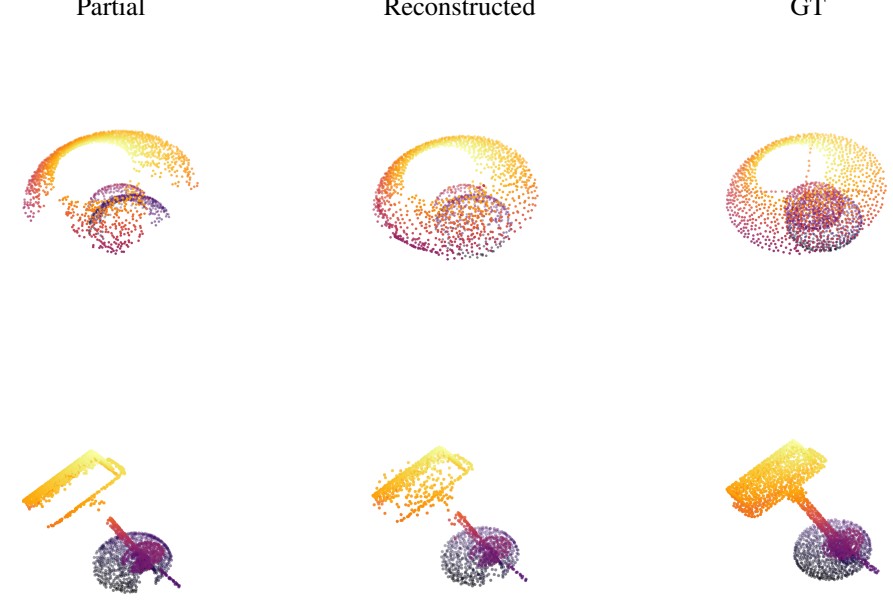

Figure 2: Qualitative visualization of difficult samples for the supervised setting.

Partial             Reconstructed             GT

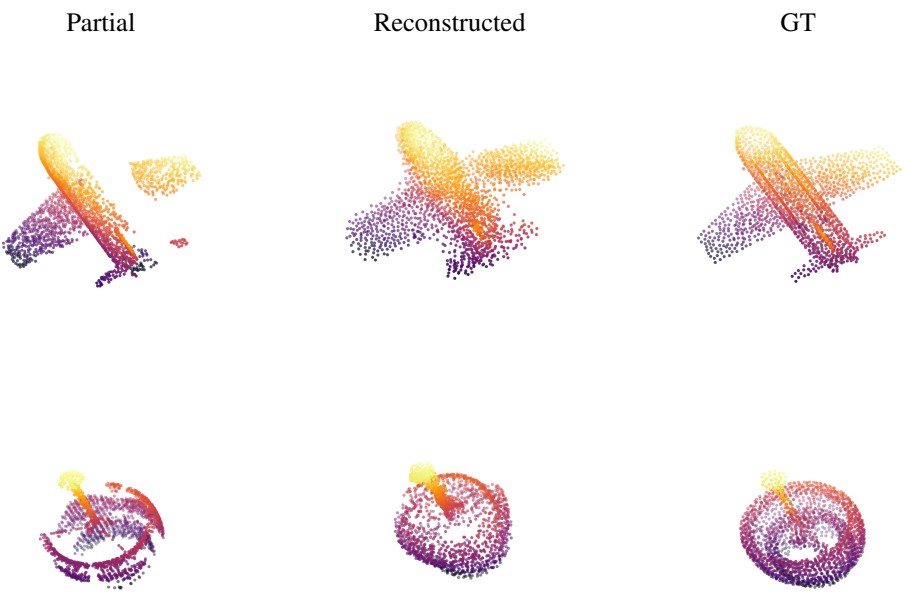

Figure 3: Qualitative visualization of difficult samples for the self-supervised setting.