# OpenReview forum: "Cross-modal Learning for Image-Guided Point Cloud Shape Completion"
_NeurIPS.cc/2022/Conference — NeurIPS 2022 Accept_

### Official Review · Reviewer_eDcw · 2022-06-28

**Rating:** 5
**Confidence:** 4
**Soundness:** 3 good
**Presentation:** 3 good
**Contribution:** 3 good

**Summary:**

The paper proposes a point cloud completion approach with images as auxiliary information. The main challenge while utilizing the information from image for point cloud competition is effectively fusing local context between image and partial point cloud. This paper proposes using cross attention modules to fuse the local information from the image and point cloud. Furthermore, to supervised the neural network, two losses are used: 1) supervised loss based on chamfer distance between predicted and ground truth point cloud and 2) self-supervised silhouette reconstruction loss with the help of a differentiable rendering of predicted point cloud.
Experiments are done on ShapeNet-ViPC and the proposed approach outperforms the baselines.


**Questions:**

1. Explain M, N, and F in Figure 1. The figure and its captions should be self-contained. Try to have consistent notations in Eq 3,4 and the Figure 1.
2. What is \beta in the Eq. 8?
3. How are hyper parameters selected? Is there is a validation set?
4. To make a more convincing argument, the Figure 3 can be modified to show an average over several shapes.



**Limitations:**

I think more qualitative results indicating failure cases should be provided.

**Strengths And Weaknesses:**

**Strengths:**
1. The problem statement is clear, contributions are clearly highlighted and the method section is well written.
2. Fusion of local information between two modalities with the help of cross attention is a good idea considering that the point cloud is a partial representation of the original shape and direct one-to-one mapping between two modalities doesn't exist.
3. Interestingly the differentiable rendering of the silhouette gives rather a large improvement even though the silhouette is a rather weak signal.

**Weakness**
1. The main limitation of the work is its benchmarking, which is done on ShapeNet shapes. Though mentioned by the authors are the limitation, it still makes the paper rather weak because we don't know how well it works in complicated scenes (ScanNet).
2. The loss function is not fully ablated for example density aware Chamfer Distance.
3. Error bars are not shown in the experiments. I understand that the benchmark shows the best performing result, still, error bars can be shown in a separate table or in the Supplementary material.
4. Why do experiments in Table 1 only show 8 categories whereas the original dataset consists of 13 categories?
5. The mixup loss only provides a tiny improvement, is it worth including it in the training? How much computational overhead this does loss contribute to the training?

---

> ### Author Response · Authors · 2022-07-29
> **Response to reviewer eDcw**
>
> We thank the reviewer for the constructive feedback and for all the suggestions to improve the paper.
>
> > *The main limitation of the work is its benchmarking, which is done on ShapeNet shapes. Though mentioned by the authors are the limitation, it still makes the paper rather weak because we don't know how well it works in complicated scenes (ScanNet).*
>
> For what concerns limitations, it would be interesting to extend our study to a real-world context, which is undoubtedly more sophisticated and articulated, but it would be a non-trivial extension due to the presence of various additional aspects discussed below. Thus, having a preliminary step in which the problem is analyzed in a controlled setting is a good and routine practice. Indeed, the vast majority of papers related to point cloud completion focus on synthetic datasets, and the state of the art for multimodal completion (ViPC) did not consider real datasets for benchmarking. Furthermore, real scenes often introduce unique issues that may necessitate ad hoc changes to the architecture and proper design decisions; for these reasons, applying the proposed framework to scenes has been left for future work. Moreover, real datasets such as ScanNet or KITTI shapes do not have ground truth complete point clouds and the evaluation is qualitative, sometimes performed with a subjective study as in Pan et al. “Variational Relational Point Completion Network”. Regarding the potential negative societal impacts of our study, we do not see a direct connection towards malicious usage of the shape completion method presented in the paper.
>
> > *The loss function is not fully ablated for example density aware Chamfer Distance.*
>
> We omitted this ablation due to space constraints, but we will include it in the final paper. We added it in the revised supplementary material (Section 5) where we also report a qualitative visualization to show the effectiveness of DCD at improving visual quality with more uniform point distributions.
>
> >  *Error bars are not shown in the experiments. I understand that the benchmark shows the best performing result, still, error bars can be shown in a separate table or in the Supplementary material.*
>
> We provide error bars for each category in the revised supplementary material (Section 7).
>
> >  *Why do experiments in Table 1 only show 8 categories whereas the original dataset consists of 13 categories?*
>
> In order to ensure a fair comparison with published results, we followed the setting of ViPC that only uses 8 categories.
>
> > *The mixup loss only provides a tiny improvement, is it worth including it in the training? How much computational overhead this does loss contribute to the training?*
>
> The mixup loss provides only a small improvement from the perspective of the Chamfer Distance. However it substantially improves the completed shape from a qualitative point of view, helping the network generate more complete shapes. Moreover, the computational overhead due to creating the mixed input shape is very small, so we decided to keep the method as it offered a very favorable cost-performance tradeoff.
>
> > *Explain M, N, and F in Figure 1. The figure and its captions should be self-contained. Try to have consistent notations in Eq 3,4 and the Figure 1.*
>
> N is the number of points of the input point cloud; M is the number of points generated by each branch of the decoder, while F represents the feature dimension. We will include these details in the caption in the revised paper.
>
> > *What is \beta in the Eq. 8?*
>
> \beta is the weight of the weighted Chamfer Distance and it is applied to penalize the distance from each point in  \hat{Y} to its nearest neighbor in Y since \hat{Y} consists in a complete point cloud while Y is the partial point cloud used as a pseudo ground truth. This will be clarified in the final paper.
>
> > *How are hyper parameters selected? Is there is a validation set?*
>
> We used a subset of the training set as a validation set, and we performed a small amount of cross validation for selecting the hyperparameters. However, for the final models we included the validation set in the training dataset.
>
> > *To make a more convincing argument, the Figure 3 can be modified to show an average over several shapes.*
>
> We appreciate the suggestion and we will modify Figure 3 in the final version showing an average over several shapes of the same category.
>
> Finally, we included in the supplementary material some failure cases where the model struggles to reconstruct perfectly.

---

### Official Review · Reviewer_WuT7 · 2022-07-10

**Rating:** 5
**Confidence:** 5
**Soundness:** 2 fair
**Presentation:** 3 good
**Contribution:** 2 fair

**Summary:**

The authors propose a new method for point cloud completion from the depth and RGB images. With the proposed cross attention and self-attention, the RGB and depth information can be better leveraged. Experimental results indicate that the proposed method outperforms the state-of-the-art methods. The self-supervised approach also outperforms a number of supervised methods.

**Questions:**

Please refer to the weaknesses.

**Limitations:**

In the Introduction, the authors claim that the application of the proposed method is to complete the missing shape of an object due to the wide use of RGBD cameras. However, the manuscript does not provide any experiments on real-world scenarios.

**Strengths And Weaknesses:**

**Strengths**

- The paper is clearly written.
- The experimental results are solid in both supervised learning and self-supervised learning.

**Weaknesses**

- The running time and model size comparison should be included.
- Most of the competitive methods only take the point cloud as input. Adding the RGB information will definitely improve the completion results. The proposed method outperforms ViPC significantly, but ViPC is not well compared to the other methods. PSGN (Fan et al., CVPR'17) should also be regarded as a simple baseline.
- Maybe it is harsh, but the manuscript does not bring new technical contributions.

---

> ### Author Response · Authors · 2022-07-29
> **Response to reviewer WuT7**
>
> We thank the reviewer for the constructive feedback and for all the suggestions to improve the paper. We address the different highlighted weaknesses separately:
>
> > *The running time and model size comparison should be included.*
>
> We agree that model size comparison and running time should be reported and they will be included in the final paper. For the time being, we have included it in the supplementary material (Section 6). Our model has a lower number of parameters with respect to ViPC and VRC and is also faster in inference due to the efficient decoder structure.
>
> > *Most of the competitive methods only take the point cloud as input. Adding the RGB information will definitely improve the completion results. The proposed method outperforms ViPC significantly, but ViPC is not well compared to the other methods. PSGN (Fan et al., CVPR'17) should also be regarded as a simple baseline.*
>
> Single view reconstruction methods like PSGN only take an image as input, thus being at a disadvantage with respect to the multimodal completion methods like ViPC and XMFNet that also exploit a partial point cloud. In their original paper (Zhang et al. , “View Guided Point Cloud Completion '', CVPR’21), the authors compare it against the suggested PSGN method, concluding that ViPC achieves superior performance. Indeed, the overall Chamfer Distance of PSGN on ShapeNet-ViPC is 7.092, the one of ViPC is 3.308 and ours is 1.443. For this reason we did not include comparisons against single-view reconstruction baselines.
>
> > *Maybe it is harsh, but the manuscript does not bring new technical contributions.*
>
> We think that the combination of implicit fusion at the feature level and novel design elements such as the attention-based upsampling module provide a substantial improvement with respect to the existing literature based on the single-view reconstruction and explicit fusion paradigm. Moreover, we are the first to propose a self-supervised setting for multimodal completion that could be useful when complete 3d scans are unavailable; this contribution has been particularly appreciated by other reviewers.
>
> For what concerns limitations, it would be interesting to extend our study to a real-world context, which is undoubtedly more sophisticated and articulated, but it would be a non-trivial extension due to the presence of various additional aspects discussed below. Thus, having a preliminary step in which the problem is analyzed in a controlled setting is a good and routine practice. Indeed, the vast majority of papers related to point cloud completion focus on synthetic datasets, and the state of the art for multimodal completion (ViPC) did not consider real datasets for benchmarking. Furthermore, real scenes often introduce unique issues that may necessitate ad hoc changes to the architecture and proper design decisions; for these reasons, applying the proposed framework to scenes has been left for future work. Moreover, real datasets such as ScanNet or KITTI shapes do not have ground truth complete point clouds and the evaluation is qualitative, sometimes performed with a subjective study as in Pan et al. “Variational Relational Point Completion Network”. Regarding the potential negative societal impacts of our study, we do not see a direct connection towards malicious usage of the shape completion method presented in the paper.

---

> > ### Comment · Reviewer_WuT7 · 2022-08-06
> > **Final Decision**
> >
> > The rebuttal addressed my concerns. Therefore, I change my ratings to borderline accept.

---

### Official Review · Reviewer_jHpu · 2022-07-10

**Rating:** 5
**Confidence:** 2
**Soundness:** 3 good
**Presentation:** 3 good
**Contribution:** 2 fair

**Summary:**

This method proposes a corss-modal learning for image-guided point cloud shape completion. In this work, they employ a mixed strategy to train the network in a supervised and self-supervised manner.

**Questions:**

I have several conerns for this work:
1. In the introduction, their novel method is proposed to address one limitation in ViPC. However, the strategy used in ViPC is explicit, while the proposed method is implicit. Actually, once the fusion strategy happens in the feature domain, it is difficult to say what happens during the fusion process. If the authors want to say that their proposed method significantly outperforms ViPC, they should better clarify the limitations of ViPC.
2. In the checklist, the authors say that they have disccused the limitations in their manuscript, while it is actually just a hand-wavy sentence in the conclusion. I do not think this is a serious discussion for the limitation of this work.
3. What is the main advantage of the proposed self-supervised loss. It seems that it is a very important component in this framework. However, I cannot find the main contribution of this module, considering the existence of the reconstruction loss. Besides, in the ablation study, the authors also say that this module just allows the training process to have a faster and smoother convergence. This description makes the module less important. Is is possible to provide more results like Figure 5?
4. In the third column of Figure 4, the results obtained by XMFNet cannot outperforms PCN and VRCnet.

**Ethics Review Area:**

["I don’t know"]

**Limitations:**

I have shown all my concerns in above sections.  I have read the rebuttal and keep my original rating. Thanks.

**Strengths And Weaknesses:**

The image-guided framework seems to be an interesting setting for this problem.  On the other hand, the use of self-supervised stratgey also increases the generalization and robustness of this framework.

---

> ### Author Response · Authors · 2022-07-29
> **Response to reviewer jHpu**
>
> We thank the reviewer for the constructive feedback and for all the suggestions to improve the paper.
> We address the different highlighted weaknesses separately:
>
> > *1. In the introduction, their novel method is proposed to address one limitation in ViPC. However, the strategy used in ViPC is explicit, while the proposed method is implicit. Actually, once the fusion strategy happens in the feature domain, it is difficult to say what happens during the fusion process. If the authors want to say that their proposed method significantly outperforms ViPC, they should better clarify the limitations of ViPC.*
>
> ViPC is bottlenecked by the need to reconstruct a coarse point cloud from the image using single view reconstruction techniques, thus performing the fusion explicitly; this is a generally hard inverse problem in itself, and full of pitfalls. Our findings show that their approach is sub-optimal and implicit fusion achieves better performance. The claim “the proposed method significantly outperforms ViPC” is supported by the strong experimental results obtained in the experiments.
>
> >*2.  In the checklist, the authors say that they have disccused the limitations in their manuscript, while it is actually just a hand-wavy sentence in the conclusion. I do not think this is a serious discussion for the limitation of this work.*
>
> We agree with the reviewer that we have not discussed limitations at length. Indeed, our major limitation is the lack of study of a real-world scenario for the proposed framework. It would be interesting to extend our study to a real-world context, which is undoubtedly more sophisticated and articulated, but it would be a non-trivial extension due to the presence of various additional aspects discussed below. Thus, having a preliminary step in which the problem is analyzed in a controlled setting is a good and routine practice. Indeed, the vast majority of papers related to point cloud completion focus on synthetic datasets, and the state of the art for multimodal completion (ViPC) did not consider real datasets for benchmarking. Furthermore, scenes introduce unique issues that may necessitate ad hoc changes to the architecture and proper design decisions; for these reasons, applying the proposed framework to scenes has been left for future works.  Moreover, real datasets such as ScanNet or KITTI shapes do not have ground truth complete point clouds and the evaluation is qualitative, sometimes performed with a user study as in Pan et al. “Variational Relational Point Completion Network”. We will include an extensive discussion of the limitations in the final version of the paper.
>
> > *3. What is the main advantage of the proposed self-supervised loss. It seems that it is a very important component in this framework. However, I cannot find the main contribution of this module, considering the existence of the reconstruction loss. Besides, in the ablation study, the authors also say that this module just allows the training process to have a faster and smoother convergence. This description makes the module less important. Is is possible to provide more results like Figure 5?*
>
> We would like to clarify that our work considers two separate settings: i) the supervised setting in which one has access to complete points and just employs the supervised reconstruction loss; ii) the self-supervised one (or weakly-supervised following the comment of reviewer dnf9) for which complete shapes are not available and where we propose our self-supervised loss combining partial reconstruction, mixup and rendering for pseudo-supervision via the auxiliary image. The two approaches are not mixed together. Indeed, when a complete ground truth point cloud is available, as in the supervised setting, the reconstruction loss using the complete shape is extremely powerful, and the contribution of a mixed strategy with the rendering fidelity term does not increase performance. However, in circumstances where 3D ground truths are not available, as in the self-supervised setting that we propose, the weaker supervision provided by the rendering loss becomes extremely valuable, allowing our method to approach the performance of supervised methods despite our lack of complete shapes.
>
> > *4. In the third column of Figure 4, the results obtained by XMFNet cannot outperforms PCN and VRCnet.*
>
> It is true that in Table 3 (we assume the reviewer intended to reference Table 3 instead of Fig.4) the self-supervised XMFNet cannot outperform the supervised counterpart of PCN and VRCNet. However, those are baselines trained in a supervised manner with access to complete shapes, while we are testing our proposed method in the self-supervised approach. It is thus remarkable that despite the lack of ground truth, the proposed method is able to approach the results of those supervised baselines and sometimes outperform them.

---

### Official Review · Reviewer_dnf9 · 2022-07-10

**Rating:** 5
**Confidence:** 4
**Soundness:** 3 good
**Presentation:** 4 excellent
**Contribution:** 2 fair

**Summary:**

This paper tackles the problem of point cloud completion through using an incomplete point cloud and an auxiliary view-image as input. Moreover, this paper proposes using view-image as supervised signal and conducting self-supervised point cloud completion.

**Questions:**

See the weakness.

**Limitations:**

The authors have not addressed the limitations and potential negative societal impact of this work.

**Strengths And Weaknesses:**

Strengths:
(1) This paper achieves state-of-the-art results on point cloud completion. (2) Furthermore, it is the first proposing self-supervised setting for point cloud completion task, and I think it is the core contribution of this paper. Also, by only using the self-supervised setting, this paper achieves a comparable result with previous fully-supervised manners. (3) The writing for this paper is relatively straightforward and well-organized.

Weakness:
 (1) This paper is heavily based on the previous work ViPC. The core differences in the fully-supervised setting are only cross-modal fusion (attention), which is widely used for other multi-modalities tasks.
(2) The Unimodal result in Table 3 is already higher than that of previous art (multi-modal), however, there is no clear illustration of why baseline performs pretty well. In addition, the performance improvement through introducing an auxiliary view-image seems marginal.
(3) I think the self-supervised method in this paper seems overclaimed. Since auxiliary view-image is introduced in the supervision which contains partial information for the point cloud ground truth, it seems more like a weakly-supervised setting.
 (4) There is no ablation for the DCD loss.

---

> ### Author Response · Authors · 2022-07-29
> **Response to reviewer dnf9**
>
> We thank the reviewer for the constructive feedback and for all the suggestions to improve the paper.
> We address the different highlighted weaknesses separately:
>
> >*(1) This paper is heavily based on the previous work ViPC. The core differences in the fully-supervised setting are only cross-modal fusion (attention), which is widely used for other multi-modalities tasks.*
>
> It is true that the cross-modal attention has been used in the literature for other multi-modality tasks, however we are the first to propose its usage in this specific setting, where the irregular domain of a 3D point cloud and an image constitute our modalities. We believe that the shape completion task could be advanced by taking advantage of the cross-attention. Moreover, as we underline in the next point, besides the fusion module we introduce some other novel elements in the design of the completion model.
>
> >*(2) The Unimodal result in Table 3 is already higher than that of previous art (multi-modal), however, there is no clear illustration of why baseline performs pretty well. In addition, the performance improvement through introducing an auxiliary view-image seems marginal.*
>
> The fact that our unimodal baseline already outperforms the previous multimodal state of the art underlines the limitations of the previous approach (ViPC). Our unimodal architecture's effectiveness is primarily due to our powerful attention-based upsampling module, combined with our encoder's local feature extraction abilities and the stacked self-attention layers for point cloud features, which model long-range relationships between local features at various levels.
> While the improvement provided by the auxiliary image may seem small in the “average” Chamfer Distance results, we want to underline that this average is performed over a large number of views, many of which presenting similar occlusions to the partial point point, thus failing to provide substantial extra information. When a good view is available, the improvement can be substantial (refer to Fig.3 for an example).
>
> > *(3) I think the self-supervised method in this paper seems overclaimed. Since auxiliary view-image is introduced in the supervision which contains partial information for the point cloud ground truth, it seems more like a weakly-supervised setting.*
>
> We termed the setting as self-supervised since we do not possess the ground truth  complete 3d shape as supervision. In the unimodal point cloud literature, this situation is commonly referred to as self-supervised. However, in the suggested multimodal context, we agree with the reviewer that a form of ground truth is actually available, albeit as a degraded 2D projection. As a result, we recognize that it is preferable to refer to the proposed solution as weakly-supervised, and we will change this in the final version of the paper.
>
> > *4) There is no ablation for the DCD loss.*
>
> We omitted this ablation due to space constraints, but we will include it in the final paper. We added it in the supplementary material (Section 5) where we also report a qualitative visualization to show the effectiveness of DCD at improving visual quality with more uniform shapes.
>
> >*Limitations*
>
> The main limitation of our work is the lack of study of a real-world scenario for the proposed framework. Indeed, it would be interesting to extend our study to a real-world context, which is undoubtedly more sophisticated and articulated, but it would be a non-trivial extension due to the presence of various additional aspects discussed below. Thus, having a preliminary step in which the problem is analyzed in a controlled setting is a good and routine practice. Indeed, the vast majority of papers related to point cloud completion focus on synthetic datasets, and the state of the art for multimodal completion (ViPC) did not consider real datasets for benchmarking. Furthermore, real scenes often introduce unique issues that may necessitate ad hoc changes to the architecture and proper design decisions; for these reasons, applying the proposed framework to scenes has been left for future work. Moreover, real datasets such as ScanNet or KITTI shapes do not have ground truth complete point clouds and the evaluation is qualitative, sometimes performed with a subjective study as in Pan et al. “Variational Relational Point Completion Network”. Regarding the potential negative societal impacts of our study, we do not see a direct connection towards malicious usage of the shape completion method presented in the paper.

---

> > ### Comment · Reviewer_dnf9 · 2022-08-08
> > **Final Decision**
> >
> > The authors addressed most of my concerns.  I will change the final score to borderline accept.

---

### Meta-Review · Area_Chair_JqWu · 2022-08-26

**Recommendation:** Accept
**Confidence:** Certain

**Metareview:**

The paper proposes a point cloud completion method that can take an auxiliary image as guidance. All the reviewers rate the paper slightly above the bar. They like the reported strong performance over the prior baseline and also the capability of using the auxiliary input. Although several reviewers raise concerns about missing experiments on real datasets such as ScanNet or KITTI, they still think the paper has sufficient merit. The AC finds no strong reason to disagree with the reviewers.

**Award:**

No

---

### Decision · Program_Chairs · 2022-09-14

Accept